# Regulation Is in the Air: The Relationship between Hypoxia and Epigenetics in Cancer

**DOI:** 10.3390/cells8040300

**Published:** 2019-04-01

**Authors:** Diego Camuzi, Ísis Salviano Soares de Amorim, Luis Felipe Ribeiro Pinto, Leonardo Oliveira Trivilin, André Luiz Mencalha, Sheila Coelho Soares Lima

**Affiliations:** 1Programa de Carcinogênese Molecular, Instituto Nacional de Câncer, Rio de Janeiro CEP 20231-050, Brazil; drdcamuzi@gmail.com (D.C.); lfrpinto@inca.gov.br (L.F.R.P.); 2Laboratório de Biologia do Câncer (LABICAN), Departamento de Biofisica e Biometria (DBB), Instituto de Biologia Roberto Alcântara Gomes (IBRAG), Universidade do Estado do Rio de Janeiro (UERJ), Rio de Janeiro CEP 20511-010, Brazil; isisssoares@yahoo.com (Í.S.S.d.A.); almencalha@yahoo.com.br (A.L.M.); 3Programa de Pós-Graduação em Ciências Veterinárias, Universidade Federal do Espírito Santo (UFES), Espírito Santo CEP 29500-000, Brazil; leotrivilin@gmail.com

**Keywords:** hypoxia, hypoxia-inducible factors, epigenetics, DNA methylation, histones modifications, cancer hallmarks

## Abstract

Hypoxia is an inherent condition of tumors and contributes to cancer development and progression. Hypoxia-inducible factors (HIFs) are the major transcription factors involved in response to low O_2_ levels, orchestrating the expression of hundreds of genes involved in cancer hallmarks’ acquisition and modulation of epigenetic mechanisms. Epigenetics refers to inheritable mechanisms responsible for regulating gene expression, including genes involved in the hypoxia response, without altering the sequence of DNA bases. The main epigenetic mechanisms are DNA methylation, non-coding RNAs, and histone modifications. These mechanisms are highly influenced by cell microenvironment, such as O_2_ levels. The balance and interaction between these pathways is essential for homeostasis and is directly linked to cellular metabolism. Some of the major players in the regulation of HIFs, such as prolyl hydroxylases, DNA methylation regulators, and histone modifiers require oxygen as a substrate, or have metabolic intermediates as cofactors, whose levels are altered during hypoxia. Furthermore, during pathological hypoxia, HIFs’ targets as well as alterations in epigenetic patterns impact several pathways linked to tumorigenesis, such as proliferation and apoptosis, among other hallmarks. Therefore, this review aims to elucidate the intricate relationship between hypoxia and epigenetic mechanisms, and its crucial impact on the acquisition of cancer hallmarks.

## 1. Microenvironment in Solid Tumors: Hypoxia

Hypoxia is a common feature of solid tumors, associated with tumor progression, resistance to treatment, and poor prognosis. The microenvironment of low oxygen levels arises as a consequence of the tumor cells high proliferation rate, which is often not accompanied by an efficient angiogenesis [1,2,3,4].

Cell response to hypoxia is mainly regulated by hypoxia-inducible factors (HIFs), a family of transcription factors involved in coordinating the expression of many genes that allow adaptation of the tumor cell to this hostile environment [2,5,6]. Hypoxia-inducible factors are heterodimeric transcription factors, which can be constituted by one of the three α-subunits that are regulated by oxygen (HIF-1α, HIF-2α and HIF-3α), and a constitutive subunit HIF-1β (also known as aryl hydrocarbon receptor nuclear translocator, ARNT) [7,8,9].

The two types of HIF subunits are classified as basic helix-loop-helix (HLH)-PER-ARNT-SIM (bHLH-PAS) proteins [10]. Alpha subunits are constituted by a DNA-binding domain, which is the basic HLH domain; and two PAS domains (PAS-A and PAS-B), which mediate the interaction with the subunit HIF-1β. In addition, the N-terminal transactivation domain (N-TAD), found in the oxygen-dependent degradation domain (ODD), where hydroxylation occurs, together with the c-terminal transactivation domain (C-TAD), regulate hypoxia-inducible factors transcriptional activity [11,12,13]. HIF-1α and HIF-2α have both N-terminal and C-terminal TADs, while HIF-3α has only the N-terminal TAD [14].

HIF-1α and HIF-2α are the most commonly studied subunits, whereas the HIF-1α is the first subunit to respond to hypoxia, and HIF-2α is stabilized after longer hypoxia periods [15]. Although these two subunits share redundant functions, they also exhibit unique and even opposing activities [6]. For example, by altering the activity of transcription factors such as p53 and MYC, HIF-1α and HIF-2α exert different effects on cell proliferation and apoptosis, with HIF-1α being known to inhibit proliferation and induce apoptosis, whereas HIF- 2α has the opposite effect [6].

On the other hand, HIF-3α has been described with distinct roles in hypoxia, generally thought to impact negatively by inhibiting HIF-1α and HIF2α. However, it has been suggested that HIF-3α variants play different roles in regulating gene expression, with stimulating or inhibiting effects depending on stimulus, organism and cellular source [16].

Under normoxic conditions, the hypoxia pathway is inactivated by post-translational modifications. So, in the presence of normal oxygen levels, α subunits proline residues 402 and 564 (HIF-1α), 405 and 531 (HIF-2α), and 490 (HIF-3α) are hydroxylated by the prolyl hydroxylase enzymes (PHDs). Proline modifications are then recognized by Von Hippel–Lindau (VHL) proteins that are substrates for the E3 ubiquitination complex. Thus, α subunits are rapidly degraded by the proteolytic pathway [17,18].

In hypoxia, hydroxylation rates are reduced, and α subunits are stabilized. They translocate to the nucleus, dimerize with HIF-1β/ARNT, recruit co-activators such as P300/CBP, and bind to hypoxia response elements (HRE) in the DNA consensus sequence [A/G]CGTG localized in the promoter region of target genes [19,20]. 

Hypoxia-inducible factors’ target genes allow tumor cell survival in the hypoxic microenvironment and are involved in regulating pH, angiogenesis, metabolic reprogramming, proliferation, autophagy, apoptosis, redox homeostasis, inflammation, tumor stem cell maintenance, invasion, metastasis and treatment resistance [6,21,22,23]. For instance, HIF-1α and/or HIF-2α induce the expression of genes such as *TGFA* and *CCND1* to stimulate cell cycle progression and proliferation; *BNIP3* to induce apoptosis and autophagy [24]; *GLUT1*, *HK1*, *HK2*, *PFK*, *ALDA*, *PGK1*, and *LDHA* to depict glycolysis; *ADM1* and *VEGF* to promote angiogenesis [25]; *MDR1* [26] and *ABCG2* [27] to confer treatment resistance; and *TWIST* [28] and *MMP2* [29] to stimulate invasion and metastasis.

Apart from their direct effect on gene expression, hypoxia inducible factors and hypoxia microenvironment are also capable of regulating epigenetic mediators. The term “Epigenetics” states for inheritable molecular alterations that affect gene expression without changing the bases in the DNA sequence. However, a broader definition, proposed in the 1940s by Conrad Waddington, emphasizes the importance of the interaction between environment and genes through epigenetic mechanisms to define phenotype [30]. This concept is of major importance in the interplay between hypoxia and epigenetic alterations. In fact, epigenetic mechanisms could be considered as sensors of cell exposure and as effectors of cell fate. Therefore, hypoxia consequences could be sensed by epigenetic mediators and, depending on how these responses are orchestrated, could push cells towards a transformed state. So, cancer hallmarks may be acquired by hypoxia-associated epigenetic alterations, and this association will be the focus of this review, particularly DNA methylation and histone modifications (Table 1).

## 2. Hypoxia and DNA Methylation

Cytosine DNA methylation is probably the most commonly studied epigenetic mechanism, usually associated with gene silencing. It consists of the addition of a methyl group to the carbon five of cytosines followed by guanines (CpG sites), reaction catalyzed by DNA methyl-transferases (DNMTs), which depend on the universal methyl donor S-adenosylmethionine (SAM) (Figure 1) [67]. CpG sites are spread along the whole genome, but in some specific regions, such as gene promoters and repetitive elements, a high frequency of these dinucleotides is found, characterizing the CpG islands. In general, DNA methylation in these regions is associated with transcription silencing, either by the recruitment of other proteins, as methyl-binding domain proteins (MBDs) and histone modifiers, or by the inhibition of transcription factors binding to DNA [68,69].

However, DNA methylation is a dynamic process and methyl marks can be removed by both passive and active demethylation pathways. In the first case, DNA methylation is lost during replication because of an ineffective copy of methyl marks into the newly synthesized strand. The active pathway (Figure 1), by comparison, is dependent on an orchestrated sequence of oxidation or deamination enzymatic reactions. Ten-Eleven Translocation (TETs) enzymes, dependent on alpha-ketoglutarate and oxygen, are the major players in the oxidative pathway and catalyze the conversion of 5-methylcytosine to 5-hydroxymethylcytosine, which can be further oxidized, forming 5-formylcytosine and 5-carboxylcytosine. The last two bases are not recognized as normal by the base excision repair (BER) machinery, are removed from the DNA strand, and finally replaced by a non-methylated cytosine [67,70].

The deamination pathway is controlled by the AID/APOBEC (Activity Induced Deaminase/Apolipoprotein B mRNA Editing Catalytic Polypeptide-like) family of enzymes that can deaminate 5-methylcytosine and 5-hydroxymethylcytosine to form thymine and 5-hydroxymethyluracil, respectively, generating a mispairing between these bases and guanine. This is recognized by the BER machinery, which removes the mispaired base to bring back a non-methylated cytosine [67,70].

Whereas in normal cells promoter-associated CpG islands are usually demethylated and repetitive regions are hypermethylated (and consequently silenced), the inverse profile is observed in tumor cells, with global genomic hypomethylation and hypermethylation of tumor suppressors, which contributes to both carcinogenesis and tumor progression. In solid tumors, hypoxia may be able to promote this aberrant DNA methylation profile [31,71]. Thienpont et al. [31] showed that oxygen shortage leads to a reduced TET activity (without decrease in *TET* expression), resulting in tumor suppressor gene promoter hypermethylation. Furthermore, the hypermethylated genes were functionally grouped into cell cycle arrest, DNA repair and apoptosis, as well as suppressor genes of glycolysis, angiogenesis and metastasis.

Hypoxia-inducible factors can also act directly on the regulation of DNA methylation mechanism. Liu and colleagues [32] showed that hypoxia reduces the level of the universal methyl donor, S-adenosylmethionine (SAM), causing hypomethylation of genomic DNA in hepatoma cells. Although the mechanisms are not clear, the authors observed that hypoxia induces *MAT2A* (methionine adenosyltransferase) expression through HIF-1α, which results in the increase of the enzyme activity and a decrease in SAM production, thus inducing genomic DNA demethylation. This is intriguing since MAT2A is one of the enzymes responsible for SAM synthesis and, therefore, the induction of its expression was expected to result in higher SAM levels. Based on this, the authors hypothesize that SAM could be consumed for polyamine biosynthesis or the different kinetic of *MAT* isoforms could play a role. Furthermore, SAM inhibits MAT2A activity [72]. Lower SAM levels have also been observed in the brain of rats after chronic cerebral hypoperfusion, although higher global methylation levels were detected [35]. In contrast, Hermes et al. [33] observed the opposite—increased SAM levels following hypoxia in HepG2 hepatoma lineage. In 2005, the same group showed that hypoxic HeLa cells present decreased SAM levels [34]. Taking these data into consideration, new studies should be performed to elucidate the effects of hypoxia on SAM levels in hepatic and other tumor cell types, since it might be cell-dependent. Notwithstanding, if it is proved that the hypoxic microenvironment can affect the levels of the methyl donor, its effects on the transcriptome will gain another layer of complexity, not only based on HIFs activity, but also on global DNA and histone methylation.

In addition, Skowronski et al. [36] observed that hypoxia and hypoglycemia cause DNMTs downregulation in human colorectal carcinoma cells, which may contribute to the pattern of low DNA methylation observed in colorectal tumors. However, Liu et al. [32] observed that hypoxia induced the expression of DNMT1 and DNMT3A in Hep3B hepatoma cells.

In human primary cardiac fibroblasts, Watson et al. [39] showed that the increased activity of *DNMT1* and *DNMT3B* promoters in hypoxia is regulated by HIF-1α. Likewise, hypoxia-induced HIF-2α induced DNMT1 expression, which promoted HIF-2α promoter hypermethylation and decreased its mRNA expression, functioning as a negative feedback mechanism for HIF-2α regulation in healthy human fetal lung fibroblasts [37].

Apart from HIFs’ effects on *DNMTs* expression, HIFs can also recruit these enzymes to specific gene regions and affect the transcription of important genes for tumorigenesis. The protein Sprouty 2 (SPRY2) is known to attenuate tyrosine kinase receptor signaling, and acts as a tumor suppressor. In addition, SPRY2 protein levels are often reduced in several cancers, such as liver, lung, prostate and breast, which is associated with poor prognosis and shorter survival [73,74,75,76]. To elucidate the mechanisms involved in SPRY2 downregulation, Gao et al. [38] employed the Hep3B hepatoma cells. This study showed that HIF-1α and HIF-2α facilitate DNA methylation mediated by DNMT1 in the regulatory region of *SPRY2*, leading to its decreased expression [38].

Mariani et al. [40] showed an increase in TET1 expression, mediated by HIF-1α, resulting mainly in the accumulation of 5-hydroxymethylcitosine in hypoxia-responsive genes, and the induction of the hypoxia response transcriptional program in neuroblastoma. These data indicated an interaction between epigenetic mechanisms and hypoxia-inducible factors in the regulation of gene expression related to hypoxia. Similarly, Lin et al. [41] indicated that hypoxia regulates DNA methylation by increasing HIF-1α-mediated *TET* expression in hepatocellular carcinoma cell, HepG2.

In addition, Wu et al. [42] indicated that hypoxia, through HIF-1α, enhanced the expression of TET1 and TET3 enzymes, leading to increased global hydroxymethylation. In this context, the authors also observed TNFα overexpression and activation of the TNFα-p38-MAPk signaling axis, which contributed to the acquisition of breast tumor-initiating cell characteristics.

However, Fischer and Miles, [43] have observed that TET2 expression was reduced by the action of HIF-1α on WM9 human metastatic melanoma cells and T98G glioblastoma cells. The same study showed that HIF-1α silencing reverted *TET2* downregulation and increased ascorbic acid-induced TET2-dependent 5-hydroxymethylation. They further suggested that the combined use of HIF-1 inhibitor and ascorbic acid may promote the re-expression of methylated tumor suppressors and may be useful in antitumor therapy.

Finally, Tsai et al. [44] suggested that hypoxia, through HIF-2α, upregulates *TET1* expression, which acts as a co-activator of HIF-1α and HIF-2α, contributing to the regulation of gene expression in response to hypoxia, including the promotion of epithelial-mesenchymal transition. Furthermore, they demonstrated that TET1, acting as a coactivator, increased the expression of the main regulator of cholesterol biosynthesis, *INSIG1* (insulin induced gene 1), mediated by HIF-2α. Knockdown of *TET1* and *INSIG1* reduced hypoxia-induced epithelial-mesenchymal transition (EMT). These data showed a new role of TET1 enzyme as a cofactor of HIF-1α and HIF-2α and an association between HIF-2α, TET1, *INSIG1* and hypoxia induced-EMT. 

Based on these data, we may conclude that the hypoxic phenotype is at least partially mediated by DNA methylation alterations, depending on both the modulation of SAM’s availability and the regulation of enzymes involved in DNA methylation and demethylation by HIFs. In the context of cancer, this link contributes to the establishment of a recurrent tumor epigenotype, involving both global DNA hypomethylation and tumor suppressor hypermethylation. Therefore, DNA methylation alterations induced by hypoxia may play a pivotal role in tumorigenesis.

## 3. Hypoxia and Histone Modifications

Histones are conserved proteins involved in DNA packing and play a central role in regulating accessibility to genomic information through chromatin remodeling. This process is coordinated by a long list of post-translational alterations, which includes acetylation, methylation, sumoylation, phosphorylation, among others, and it is crucial for the regulation of gene expression and DNA repair [77]. Different enzymes are responsible for writing and erasing these post-translational marks (Figure 1), but they can also modify non-histone proteins, such as HIFs, altering their stability. In addition, hypoxia regulates both the activity and expression of these enzymes, which could have a global impact on gene expression profiles.

Sirtuins (SIRTs) are a family of NAD^+^-dependent deacetylases that regulate transcription by inducing heterochromatin formation following the deacetylation of histone tails, but have also been shown to play a central role in the regulation of key transcription factors, functioning both as oncogenes and tumor suppressor genes [78]. For instance, the deacetylation of p53 at lysine residue 382 by SIRT1 leads to a reduction of its transcriptional activity [79]. Complementary, SIRT1 may also act as a tumor suppressor by deacetylating HIF-1α lysine 674, blocking p300 recruitment and repressing HIF-1α targets [46]. The same deacetylase has been shown to stimulate HIF-2α activity [47]. Other SIRTs have been shown to regulate HIF-1α stability. SIRT2-mediated deacetylation of HIF-1α increases its affinity for prolyl hydroxylase 2, which hydroxylates HIF, inducing its degradation by the proteasome [48]. However, the association between SIRTs and hypoxia is not a one-way street.

SIRT activity is directly dependent on NAD^+^ levels; therefore, these enzymes are considered sensors of the cellular redox state. Severe hypoxia increases NADH/NAD^+^ ratios [80], possibly as a consequence of oxidative phosphorylation (OXPHOS) inhibition. In this context, hypoxia would reduce SIRTs activity, leading to a positive feedback by increasing HIF-1α stability and activity. Thus, these deacetylases could represent one of the links between hypoxia and cancer. In agreement with this hypothesis, decreased SIRT1 levels have been observed in *BRCA1*-mutated breast cancers [81]. In addition, *SIRT2* downregulation was reported in glioma, hepatocarcinoma, esophageal and gastric adenocarcinomas [82,83], while small cell lung carcinoma [84], breast cancer and leukemia [85] present lower levels of *SIRT4*. Other SIRTs are also dysregulated in cancer (for a review, access [86]), but it is not clear what comes first during the carcinogenesis process, the inhibition of these enzymes by the hypoxia-associated reduction of NAD^+^ levels, the increased HIF-1α stability and activity resulting from SIRT downregulation, or even if other mechanisms are involved. For example, a hypoxic-like phenotype is induced during ageing, one of the strongest risk factors associated with cancer development, with a reduction of NAD^+^ levels. This leads to a reduction of SIRT1 activity, with a consequent HIF-1α stabilization [87]. Either way, evidence points to a strong association between sirtuins and metabolism, consequently hypoxia, and such association should be further explored in cancer, including their potential as new tumor therapy targets.

Other histone-modifying enzymes are also directly linked to HIF-1α protein stability and involved in the promotion of tumor growth. Kim et al. [49] have shown in an in vivo model that non-canonical post-translational modifications in HIF-1α determine the protein’s fate. Methylation of HIF-1α by SET7/9 (Histone-lysine N-methyltransferase SETD7) marks the protein for degradation, even in the nucleus. The demethylation process was catalyzed by lysine-specific demethylase 1 (LSD1). Mice carrying *HIF-1A* S28Y and R30Q mutations were shown to be resistant to HIF-1α methylation, resulting in protein stabilization. In these mice, the implantation of Lewis lung carcinoma cells showed higher tumor growth [49].

LSD1, a FAD-dependent demethylase, removes mono- and dimethyl marks from histone 3, inducing transcriptional disturbances dependent on the affected amino acid residue (lysine 4 or 9) [88]. While H3K4me and H3K4me2 are associated with active transcription, H3K9me2 is a repressive mark [89]. Apart from its role in chromatin regulation and HIF-1α demethylation, LSD1 is capable of augmenting HIF-1α stability, both in normoxia and hypoxia, via an oxygen-independent pathway, but dependent on its demethylase activity. LSD1 demethylates RACK1K271me2, which then fails to bring the Elongin C-containing E3 ubiquitin ligase complex to HIF-1α, avoiding its proteasomal degradation [50]. Therefore, in the early stages of normoxia and hypoxia, LSD1 plays a role in upregulating glycolysis via HIF-1α stabilization. Consistent with this, LSD1 upregulation is observed in hematological and solid tumors [90]. However, during prolonged hypoxia, FAD^+^ levels decrease as a consequence of the diminished expression of riboflavin kinase (RFK) and flavin adenine dinucleotide synthetase 1 (FLAD1), leading to impaired LSD1 activity and consequent HIF-1α degradation [50]. This strong correlation between LSD1-promoted demethylation and HIF-1α levels is further reinforced by observations in human samples. In triple-negative breast cancer, both hypoxia pathway signature and the average expression of LSD1 target genes are associated with a poor prognosis [50,91]. Therefore, LSD1 inhibitors are currently in clinical trials involving different tumor types, such as acute myeloid leukemia, small cell lung carcinoma, prostate cancer, among others [90].

Histone demethylases from the Jumonji Domain-Containing (JMJD) family of proteins have also been suggested to play a role in the interplay between hypoxia and epigenetics. Wellmann et al. [53] showed in human embryonic kidney cells (HEK-293) and human microvascular endothelial cells (HMEC-1) that hypoxia and hypoxia mimetic agents induce *JMJD1A* expression, which is abrogated after *HIF-1A* silencing [53]. The authors further showed that *JMJD1A* promoter harbors a hypoxia responsive element, enabling its induction by HIF-1α both in vitro and in vivo. HIF-1α is also able to bind to *JMJD2B* promoter, inducing its expression [54]. The regulation of JMJD proteins by hypoxia was also evaluated in other human cancer cell lines (U2OS, MCF7, HeLa, IMR32 and HL60), confirming the induction of *JMJD1A* and *JMJD2B* [55]. *JMJD2B* upregulation by HIF-1α during hypoxia has been linked to the modulation of hypoxic gene expression, and to increased cell proliferation [51]. Furthermore, the regulation of a subset of hypoxia-inducible genes, including *ADM* and *GDF15*, has been shown to be dependent on JMJD1A in renal cell and colon carcinoma cell lines, reinforcing the connection between hypoxia, HIF-1α and JMJD-mediated chromatin remodeling [52].

Acetyl-CoA, the fuel of tricarboxylic acid cycle (TCA) cycle, is the universal donor for acetylation reactions in a cell. Therefore, the activity of histone acetyl-transferases (HATs) is highly dependent on the concentrations of this intermediate. HATs transfer an acetyl group to histone tails, especially to lysine residues, annulling their positive charge and causing a weaker interaction with the DNA molecule, with a consequent euchromatin formation [92]. In hypoxic conditions, the stabilization of HIF-1 complex leads to activation of pyruvate dehydrogenase kinase 1 (PDK1), blocking pyruvate dehydrogenase (PDH) activity. As a consequence, acetyl-CoA synthesis from pyruvate is impaired [56]. In this context, it is reasonable to hypothesize that a global reduction of histone acetylation would take place in hypoxic microenvironments, inducing a heterochromatin state. In fact, a widespread repression of RNA and mRNA synthesis is induced by hypoxia [57], although the reduction of HAT activity cannot be pointed at as the sole cause. As previously mentioned, the activity of other histone modifiers is modulated during O_2_ deprivation by NAD^+^ and FAD^+^ altered levels and, to add more complexity to this scenery, the activity of some histone demethylases can be limited by α-ketoglutarate availability.

Alpha-ketoglutarate (or 2-oxoglutarate) is a rate-limiting intermediate of the TCA cycle, which plays an important role in cellular energy metabolism, protein synthesis modulation, immune system homeostasis, among others [93]. As previously mentioned, it is a cofactor of TET enzymes, involved in the active demethylation process, but it is also a cofactor of JMJD. During hypoxia, α-ketoglutarate and the following citrate production through oxidative metabolism of glucose is reduced, but α-ketoglutarate levels are actually increased. In these conditions, α-ketoglutarate is produced from glutamine metabolism, and it goes through reductive carboxylation to form citrate, which supports cytosolic macromolecular synthesis, enabling cell proliferation even in a hypoxic microenvironment [94]. Therefore, TETs and JMJDs could have enough α-ketoglutarate available to catalyze their reactions. However, the increase in α-ketoglutarate levels during hypoxia is followed by an increase of 2-hydroxyglutarate (2HG) production by lactate dehydrogenase A in cells with wild-type *IDH1* and *IDH2*. 2-hydroxyglutarate has been considered an oncometabolite, since it can block tumor cell differentiation, maybe because of its potential inhibition of α-ketoglutarate-dependent enzymes such as TETs and JMJDs [58,94]. Indeed, 2HG produced during hypoxia is sufficient to induce the spreading of histone repressive marks, including H3K9me3 [58].

The interplay between hypoxia, HIFs and histone modifications seems to be even more complex than that with DNA methylation. Indeed, the data gathered here show that histone modifiers are capable of modulating HIFs stability, having a direct impact on hypoxic signaling, but HIFs are also capable of modulating the expression of histone modifiers. Furthermore, metabolism intermediates, highly sensible to O_2_ levels, are cofactors of these enzymes, and HIFs may also exert its transcriptional activities through the interaction with chromatin remodelers. In cancer, the expression of different histone-modifying enzymes has been shown to be dysregulated, what can be mediated at least in part by HIFs, and has an impact on tumor phenotype and prognosis. Furthermore, some hypoxia-associated tumor transcriptional programs seem to be acquired by the cooperation of HIFs and histone modifiers. Although much is left to be clarified, these observations suggest a strong connection between hypoxia and this epigenetic mechanism in the establishment of cancer hallmarks.

## 4. Painting the Cancer Hallmarks with Epigenetics

The hypoxia phenotype triggered by the activation of hypoxia-inducible factors due to low O_2_ levels may eventually become a paradox on tissue homeostasis. Some healthy tissues keep homeostasis during hypoxia, such as in early embryonic development [95], or in the intestine of adult individuals [96]. However, along the carcinogenesis and tumor progression the consequence of low O_2_ levels is selection of more resistant clones [97], and stimulation of several phenotypes linked to cancer hallmarks, such as increased angiogenesis, proliferation, invasion, metastasis, genomic instability, and immune leakage, among others [98,99,100]. Regardless of the hallmark, epigenetic mechanisms are also altered, responding to hypoxia or interacting with hypoxia-inducible factors to modulate cell fate [101,102], as described in the next sections and summarized in Figure 2 and Table 1.

### 4.1. Unbalanced Proliferation

Cell cycle is controlled by cyclins and cyclin-dependent kinases (CDK), the latter being directly inhibited by specific inhibitors (cyclin-dependent kinase, inhibitors, CDKI). The cyclin-dependent kinase Inhibitor 2A (CDKN2A^INK4A^ or p16) is responsible for monitoring the cell during the progression of G1/S and through oncogenic stimuli leads to cell senescence [103]. Alterations of p16 are found in several tumors, both downregulation and upregulation [104], and p16 protein expression is also used as a biomarker of the oncogenic Human Papillomavirus 16 (HPV16) infection, particularly in oropharyngeal tumors [105].

In human lung fibroblasts, Raf reduces H3K27me3 repressive marks present in the *p16* gene by the activation of a *p16* locus-specific demethylase, JMJD3, causing its activation. However, even if recruited to the *p16* locus, during hypoxia, the activity of the enzyme JMJD3 is reduced by O_2_ deprivation, its substrate, resulting in increased H3K27me3 levels [59]. However, despite the increase in the repressive mark, the expression of *p16* is not affected. Other analyses pointed to a bivalent chromatin state where, in addition to the increase of the repressive trimethylation in H3K27, there is also an increase in the activation trimethylation in H3K4 by the inhibition of the demethylases JARID1A and JARID1B, also caused by the reduced O_2_ levels [59]. Thus, showing a role of bivalent chromatin in the regulation of cell cycle checkpoints during hypoxia. However, this phenotype is controversial, since *p16* can be downregulated during hypoxia in mesenchymal stem cells, pointing to a tissue-specific regulation [106]. In both cases, however, non-tumoral cell lineages were used as models, raising the question of how *p16* expression would be modulated in tumor cells in hypoxic conditions.

Proliferation pathways have been shown to be altered by DNA methylation mechanisms during hypoxia. The *c-MYC* proto-oncogene is dysregulated in a series of tumors and is linked to increased proliferation and apoptosis evasion, having hypoxia-inducible factors as partners or antagonists in several processes [107]. So, while HIF-1α antagonizes c-MYC-induced proliferation, HIF-2α increases it by potentiating the oncogenic effect of c-MYC in an in vitro model with *VHL*-deficient renal cell carcinoma [108]. One of the oncogenic mechanisms resulting from this HIF-2α and c-MYC interaction may be epigenetic changes in target suppressor genes. *PLA2R1* (*Phospholipase A2 Receptor 1*) is a potential tumor suppressor gene with a few known functions, but its increased expression has been linked to cell cycle arrest and apoptosis induction [109]. Renal cell carcinomas with *VHL* deficiency and *c-MYC* amplification exhibit HIF-2α stabilization and *PLA2R1* repression by promoter hypermethylation. In vitro treatment with the DNMT inhibitor and DNA demethylating agent 5-azadeoxycytidine restored *PLA2R1* expression [45].

In contrast, in an in vivo model of sarcoma, HIF-1α is active, whereas HIF-2α has its expression inhibited. When cells are treated with Vorinostat, an inhibitor of histone deacetylases class I and II (HDAC1-10), HIF-2α expression is restored, without changes in HIF-1α levels. In these cells, the restoration of HIF-2α plays a tumor suppressor role, inhibiting the mTORC1 pathway and promoting a decrease in cell proliferation and tumor growth [60]. These differences suggest that the role of hypoxia-inducible factors in the regulation of cell proliferation might be dependent on the tumor type.

### 4.2. Angiogenesis

The main player of angiogenesis is the vascular endothelial growth factor (VEGF), regulating the creation of new vessels and vascularization of new tissues, allowing nutrient and O_2_ supply. In tumors, when cell growth overcomes that of healthy tissues, a hypoxia microenvironment is created, and VEGF can be stimulated independently or directly by hypoxia-inducible factors [110,111]. Breast carcinoma cells in hypoxia (0.01% O_2_) exhibit increased methylation in H3K4 and acetylation in H3K9, both activation marks, in *VEGF* promoter region [61]. Furthermore, some studies with in vitro or in vivo models have shown that treatment with deacetylase inhibitors can inhibit *VEGF* expression and thus angiogenesis by restoring the expression of HIF-1α suppressors such as VHL and PHD [112,113]. 

### 4.3. Metastasis and Invasion

Hypoxia is a double-edged sword, whereas low O_2_ levels induce the death of several tumor cells, the hypoxia-resistant cells exhibit a more invasive and metastatic phenotype [97]. In melanomas, the loss of 5-hydroxymethylcytosine (5hmC), caused by decreased expression of *TETs* and of isocitrate dehydrogenases (*IDH1* and *IDH2*), the latter coding for the enzymes that produce α-ketoglutarate, is a hallmark of this tumor, particularly to differentiate between melanomas and benign melanocytic nevi [114,115].

*HIF-1α* knockdown in a metastatic melanoma cell line, is followed by an increase in TET2 gene and protein levels. These cells, when supplemented with ascorbic acid (cofactor of TETs enzymes), show higher 5hmC levels, which are also observed in glioblastoma cell lines [43]. TET2 overexpression and the consequent increase in 5hmC levels decrease melanoma cell invasion in murine models [114]. Thus, targeting HIF-1α along with ascorbic acid supplementation may be a therapeutic option.

As previously mentioned, HIF-1α increases the transcription of *JMJD1A* and *JMJD2B* in malignant cell lines, such as prostate adenocarcinoma, cervical cancer [54], and clear cell renal carcinoma [52]. Increased expression of *JMJD1A* is related to the induction of the expression of several genes linked to invasion [52]. In addition, JMJD1A nuclear expression is associated with tumor stage and lymph node metastasis in oral and oropharyngeal squamous cell carcinoma [116]. The regulation of *JMJD2B* in a HIF-1α-dependent manner during hypoxia increases the invasiveness of colorectal cancer cells by reducing the levels of the repressive mark H3K9me3 in genes like *ELF3* (*E74-like ETS transcription factor 3*) and *IFI6* (*Interferon Alpha Inducible Protein 6*) [51]. *ELF3* overexpression is involved in the metastasis process in non-small cell lung cancer [117] and its expression can be a useful biomarker of lymph node metastasis in colorectal cancer [118], while IFI6 promotes metastasis in breast cancer cells [119].

*JMJD1A* is found overexpressed in hepatocellular carcinomas and is related to higher recurrence rates [62]. During hypoxia in vitro, the expression of the epithelial marker *E-cadherin* is reduced, followed by increased levels of the mesenchymal markers *N-caderin* and *Twist*. *JMJD1A* silencing during hypoxia leads to a reduction of *N-cadherin* and *Twist*, along with increased *E-cadherin* levels [62].

### 4.4. Genomic Instability

DNA damage is readily sensed and usually corrected by the DNA repair machinery in a normal cell in order to avoid the accumulation of mutations. Among the types of damage, double-strand breaks are a major source of stress and DNA repair system impairments can result in genomic instability, including changes in nucleic acid sequences, chromosomal rearrangements or aneuploidy. BRCA1 and RAD51 are classical proteins involved in double-strand break repair through homologous recombination [120] and both proteins have high clinical relevance for breast cancer [121,122]. 

Breast carcinoma cells submitted to hypoxia (0.01% O_2_) showed a decrease in H3K4 methylation (H3K4me2,3), associated with increased transcriptional activity in *BRCA1* and *RAD51* promoter regions, and, therefore, presented a decreased expression of these genes. In these cells, it was verified whether this demethylation could be performed by the demethylases JARID1A, JARID1B and LSD1; however, only LSD1 was involved. Furthermore, the suppressor marker H3K9me3 was increased in the promoter region of *BRCA1* and *RAD51*. There was also a decrease of H3K9 acetylation, a marker present in transcriptionally active regions. Interestingly, for both genes, the demethylation of H3K4 was performed independently of HIF-1α expression [61].

MLH1 and PMS2 are tumor suppressor proteins that form a heterodimer involved in DNA mismatch repair [123]. During hypoxia, an increased expression of the histone methyltransferase G9a (G9a) is observed, which is related to global and localized increase of the suppressor mark H3K9me2. Under these conditions, higher H3K9me2 levels are detected in the promoter region of *MLH1*, decreasing its expression [63]. Interestingly, cells treated with histone deacetylase inhibitors during hypoxia have restored *MLH1* levels. Although *PMS2* mRNA expression is not altered during hypoxia, there is a drop in its protein levels. These cells with MLH1 and PMS2 impairment during hypoxia have their repair system affected, resulting in increased mutagenesis and genomic instability [64].

### 4.5. Immune Modulation

Cells from the immune system act as guardians actively looking for invaders and altered cells causing the elimination of these in order to maintain body homeostasis. On the other hand, after immune escape and tumor establishment, a subversion of the immune system occurs, altering the expression of cytokines that will work in support of the tumor [124]. During this review, we observed how hypoxia influences the epigenome of malignant cells and vice versa, now we will discuss how the same phenomenon can be observed in immune cells.

During hypoxia, the expression of the demethylases *JMJD1A*, *JMJD2B* and *JMJD2D* tends to increase in macrophages possibly as a mechanism to compensate for the low activity of these enzymes due to O_2_ deprivation. Despite their increased gene expression, an overall increase in H3K9 and H3K36 methylation is observed [65], with phenotypes similar to those observed in tumors [52]. Increased H3K9 methylation affects target cytokine genes, as *monocyte chemoattractant protein-1* (*MCP-1*, also known as *CCL2*), *CC Motif Chemokine Receptor 1* (*CCR1*) and *CC Motif Chemokine Receptor 5* (*CCR5*), reducing their gene expression [65]. Similarly, in HeLa cells, *MCP-1* gene expression is also reduced during hypoxia [66].

*MCP-1* and *IL-8* possess NF-kappaB (NF-kB) responsive elements in their promoters. However, during hypoxia, *MCP-1* expression is suppressed, while *IL-8* expression is induced. Interestingly, NF-κB interacts with both genes, but, at low O_2_ levels, NF-κB promotes *IL-8* transcription, and in *MCP-1* it acts as a repressive factor, recruiting HDAC2 to its promoter region [66]. The increase in NF-kB expression is common in several tumors and is related to increased malignancy [125], the same applies to its transcriptional target *IL-8* [126]. NF-kB also directly regulates HIF-1α and HIF-1β gene expression, resulting in upregulation of several hypoxia-like genes [127,128,129].

### 4.6. Resistance to Cell Death

In healthy tissues, the balance between proliferation, differentiation and cell death is critical to maintaining the individual’s homeostasis. Evading apoptosis is a key step during carcinogenesis and progression. The ability to bypass the mechanisms of cell death is directly related to resistance to treatment and malignancy [130,131,132].

Euchromatic histone-lysine N-methyltransferase 2 (EHMT2/G9a) regulates H3K9me2 and is found overexpressed in several tumor cell lines during hypoxia [63]. The expression of G9a confers resistance to chemotherapy [133] and the pharmacological inhibition of G9a induces apoptosis in tumor cells [134,135]. Nonetheless, the connection between G9a and hypoxia on tumor progression is poorly understood.

BCL-2 interacting protein 3 (BNIP3) is an HIF target gene, involved in apoptosis and autophagy pathways [136], and its expression is increased during hypoxia in several cell lines by direct HIF-1α activation [137]. On the contrary, in hypoxic tumor cells, the *BNIP3* promoter can be hypermethylated, resulting in its silencing regardless of HIF-1α expression. Treatment with DNA demethylating agents and, interestingly not with HDAC inhibitors, restores its expression [138]. The regulation of *BNIP3* during tumor hypoxia is poorly understood and further studies are needed to understand its causal effect.

## 5. Final Considerations

Although different studies have tried to elucidate the impact of hypoxia on epigenetic programing, much is left to be learned. Nevertheless, the epigenetic machinery is an important hypoxia sensor, and copes with hypoxia-inducible factors to depict the hypoxic phenotype. However, modulation of epigenetic mediators’ expression by these transcription factors makes evident a highly complex regulation mechanism. Indeed, the levels of many cofactors and radical donors for reactions catalyzed by epigenetic regulators are affected during hypoxia and have shown to exert a global effect on gene expression.

One of the clearest consequences of the exposure to a hypoxic environment is the metabolic switch from oxidative phosphorylation (OXPHOS) to glycolysis, and not surprisingly, one of the proposed cancer hallmarks is exactly this switch. In this context, epigenetic alterations can both play a role in the establishment of this phenotype, and be affected by it, indicating a central role of epigenetics in the acquisition of this cancer hallmark associated with hypoxia. Although epigenetic alterations have also been associated with the acquisition of other cancer hallmarks, only more recently an association with hypoxia has been proposed, as made evident by the information gathered here.

Furthermore, data presented in this review suggest hypoxia-epigenetics interaction might be tissue-specific and dependent on other factors, such as oxygen deprivation duration. Although such a highly complex interaction is observed during carcinogenesis and tumor progression, modulation of specific players such as histone modifiers presents promising results in cancer cell control in vitro. This highlights the potential of alterations of this regulatory axis to be used as biomarkers of tumor aggressiveness as well as therapeutic targets.

## Figures and Tables

**Figure 1 cells-08-00300-f001:**
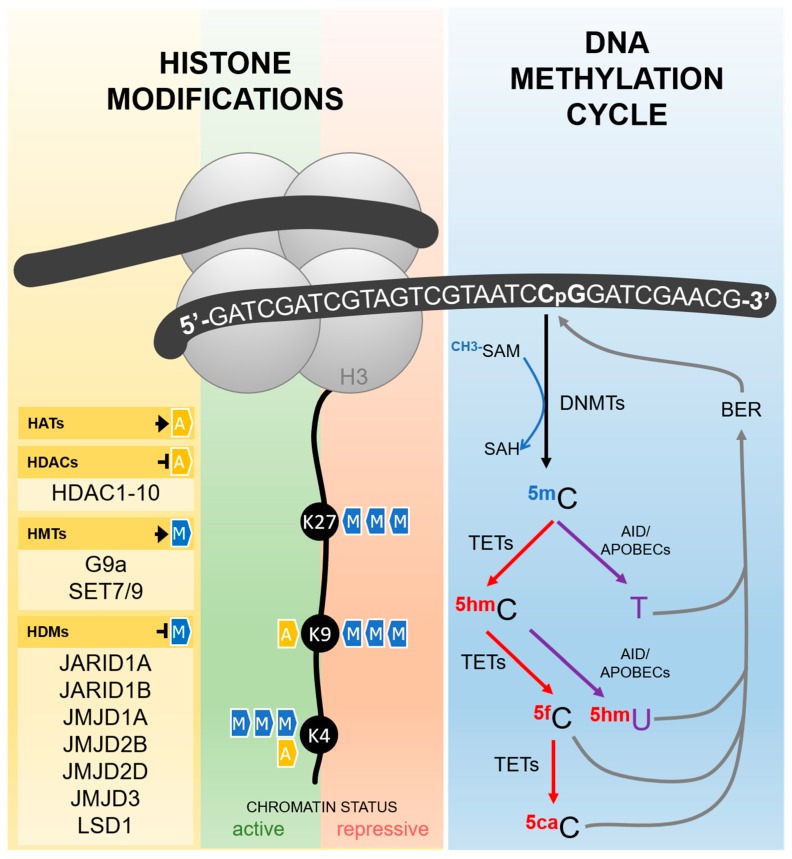
Epigenetic mediators involved in the establishment and erasure of histone post-translational modifications and DNA methylation, as described in the text. Histone acetyltransferases (HATs) are responsible for the transfer of an acetyl group to different histone amino acid residues, including lysines 4 and 9 (K4 and K9). Histone deacetylases (HDACs) catalyze the removal of this group. Methyl groups can also be added to different amino acids in the tails of the histones by histone methyltransferases (HMTs) and removed by histone demethylases (HDMs). Histone acetylation always results in an active chromatin status, while the effects of histone methylation depends on the number of groups added and the amino acid residue involved. As exemplified in the figure, H3K27me3 and H3K9me3 are repressive marks, while H3K4me3 is an active mark. DNA methylation takes place more frequently in cytosines followed by guanines in the so-called CpG sites, reaction catalyzed by DNA-methyltransferases (DNMTs) using S-adenosylmethionine (SAM) as methyl donor. The erasure of DNA methylation can be driven by an active pathway, either by sequential oxidations or deamination. In the oxidation pathway (depicted in red), Ten-eleven translocation enzymes (TETs) catalyze the conversion of 5-methylcytosine (5mC) in 5-hydroxymethylcytosine (5hmC), which is further oxidized to form 5-formylcytosine (5fC) and 5-carboxylcytosine (5caC). Both 5fC and 5caC are recognized by the base excision repair (BER) machinery and removed from the DNA strand, resulting in the reincorporation of a non-methylated cytosine. In the deamination pathway (depicted in purple), 5mC and 5hmC can be deaminated by AID/APOBEC (Activity Induced Deaminase/Apolipoprotein B mRNA Editing Catalytic Polypeptide-like) family of enzymes, generating thymine and 5-hydroxymethyluracil (5hmU), respectively. The mispairing of these bases with guanine in the opposite strand activates the BER machinery and results in the reincorporation of a non-methylated cytosine. G9a: histone methyltransferase G9a; SET7/9: histone-lysine N-methyltransferase SETD7; JARID: Jumonji/ARID Domain-Containing Protein; JMJD: Jumonji Domain-Containing Protein; LSD1: Lysine-specific demethylase 1; SAH: S -adenosylhomocysteine.

**Figure 2 cells-08-00300-f002:**
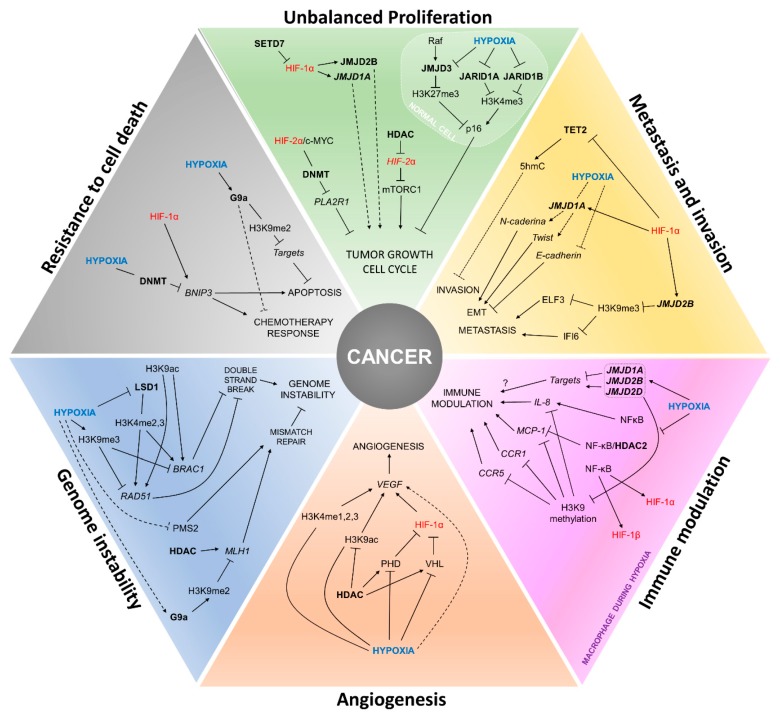
Interactions between hypoxia, HIFs and epigenetic players in the establishment of cancer hallmarks. The figure summarizes how the establishment of cancer hallmarks can be influenced by the cross-regulation of the expression and activity of HIFs and epigenetic modifiers during hypoxia. Text in blue: hypoxia due to low O_2_ levels; text in red: hypoxia-inducible factors; text in bold: epigenetic players.

**Table 1 cells-08-00300-t001:** Summary of epigenetics and hypoxia interplay.

**Hypoxia and DNA Methylation**					
**Hypoxic or Hypoxic-Like Condition**	**Epigenetic Modifier/Modification Involved**	**HIF Involved**	**Cancer/Cell Type**	**Functional Impact**	**Reference**
0.5% O_2_ for 24 h	5hmC	ND	Eleven human and murine cell lines from different normal tissues and tumor types (HepG2, HT-1080, MCF10A, H358, MCF7, Hep3B, LLC, mESC WT, N2a, mES *Tet1-/-* and A549)	Decreased 5hmC levels following hypoxia. In MCF7, 5hmC was decreased near transcription start sites of *NSD1*, *FOXA1* and *CDKN2A*	[31]
1% O_2_ for 24 h	DNA methylation	HIF-1α	Human hepatoma cells (Hep3B)	*MAT2A* induction; decreased SAM levels; genomic DNA hypomethylation	[32]
		ND	Human hepatoblastoma cells (HepG2)	Increased SAM levels	[33]
			Human cervix adenocarcinoma cells (HeLa)	Decreased SAM levels	[34]
In vivo (rats) cerebral hypoperfusion (ischemia) for 90 days	DNMT3A	ND	Brain	Decreased SAM production; higher global methylation levels; higher DNMT3A expression levels	[35]
In vitro ischemia for 24 h	DNMTs	ND	Human colorectal carcinoma cells (HCT116)	Decreased DNMTs expression, which may contribute to the low DNA methylation observed in colorectal tumors	[36]
1% O_2_ for 24 h	DNMT1 and DNMT3A	ND	Human hepatoma cells (Hep3B)	Increased DNMTs expression	[32]
2% O_2_ for up to 96 h	DNMT1	HIF-2α	Healthy human fetal lung fibroblasts (HFL1) and lung cancer	Increased *DNMT1* expression; *HIF-2**α* hypermethylation and decreased expression	[37]
3% O_2_ for 24 h		HIF-1α and HIF-2α	Human hepatoma cells (HuH7 and Hep3B)	DNMT1 recruitment to *SPRY2* promoter and its consequent decreased expression	[38]
1% O_2_ for up to 8 days	DNMT1 and DNMT3B	HIF-1α	Human primary cardiac fibroblasts (HCF)	Increased *DNMTs* expression	[39]
1% O_2_ for 48 h	TET1		Human neuroblastoma cells (SK-N-BE)	Increased TET1 expression; accumulation of 5-hydroxymethylcitosine in hypoxia-responsive genes	[40]
1% O_2_ for 24 h	TETs		Human hepatoma cells (HepG2)	Induced expression of TET enzymes	[41]
1% O_2_ for 24 h	TET1 and TET3		Human breast cancer cell lines (MCF7 and MDA-MB-231) and primary breast cancer cells	Global hydroxymethylation; TNFα overexpression and activation of the TNFα-p38-MAPk signaling axis	[42]
Normoxia	TET2		Human metastatic melanoma cells (WM9) and human glioblastoma cells (T98G)	Reduced TET2 expression	[43]
1% O_2_ for 18 h	TET1	HIF-1α and HIF-2α	Human hypopharynx carcinoma cells (FaDu) and human non-small cell lung cancer cells (derived from lymph node metastasis, H1299)	Increased TET1 expression; regulation of gene expression in response to hypoxia; *INSIG1* induced expression; promotion of epithelial-mesenchymal transition	[44]
*VHL* deficiency + *c-MYC* amplification	DNA methylation	HIF-2α	Human kidney cancer cells (ACHN, RCC10 and 786-O)	HIF-2α stabilization and *PLA2R1* repression by promoter hypermethylation	[45]
**Hypoxia and Histone Modifications**					
**Hypoxic or Hypoxic-Like Condition**	**Epigenetic Modifier/Modification Involved**	**HIF Involved**	**Cancer/Cell Type**	**Functional Impact**	**Reference**
In vitro 1% O_2_ for up to 24 h; in vivo (mice) 8% O_2_	SIRT1	HIF-1α	Human fibrosarcoma cells (HT1080), human colon cancer cells (HCT116), human embryonic kidney cells (HEK293 and HEK293T) and mice	In normoxia, SIRT1 deacetylates HIF-1α, blocking p300 recruitment, and represses HIF-1α targets. This is reversed in hypoxia, when SIRT1 levels decrease	[46]
In vitro 1% O_2_; in vivo (Sirt1+/– mice) 6% O_2_		HIF-2α	Human hepatoma cells (Hep3B) and mice	Stimulation of HIF-2α activity	[47]
1% O_2_ for 16 h	SIRT2	HIF-1α	Human cervix adenocarcinoma cells (HeLa)	Increased affinity of HIF-1α by PHD2, after deacetylation by SIRT2; HIF-1α degradation by proteasome	[48]
In vitro 1% O_2_ for up to 30 h; in vivo (mice) 10% O_2_ for up to 14 days	SET7/9 and LSD1		Human cervix adenocarcinoma cells (HeLa) and mice	Regulation of HIF-1α stability	[49]
1% O_2_ for up to 8 h	LSD1		Human embryonic kidney cells (HEK293T), human lung adenosquamous carcinoma cells (NCI-H596), human colon adenocarcinoma cells (Colo-205) and human clear cell renal cell carcinoma cells (RCC4)	Increased HIF-1α stability; glycolysis upregulation	[50]
1% O_2_ for up to 24 h			Human lung adenosquamous carcinoma cells (NCI-H596)	Decreased RFK expression, reduced FAD+ levels; HIF-1α degradation.	[50].
1% O_2_ for 24 h	JMJD2B		Human colorectal cancer cells (SW480 and HCT116)	Increased JMJD expression and proliferation induction; reduced H3K9me3 levels in ELF3 and IFI6	[51]
0.5% O_2_ for 16 h	JMJD1A		Human clear cell renal cell carcinoma cells (RCC4) and human colon cancer cells (HCT116)	Increased JMJD expression and regulation of hypoxia-inducible genes	[52]
In vitro 0.5% O_2_ or chemical hypoxia for up to 18 h; in vivo (rats) 8% O_2_ for up to 12 h			Human embryonic kidney cells (HEK293); brain, heart, kidney and liver	Increased *JMJD1A* expression	[53]
0.5% O_2_ for 16 h	JMJD1A and JMJD2B		Human prostate cancer cells (LNCaP), human cervix adenocarcinoma cells (HeLa) and human renal adenocarcinoma cells (786–0 RCC)	Increased JMJD expression	[54]
			Human osteosarcoma cells (U2OS), human breast cancer cells (MCF-7), human cervix adenocarcinoma cells (HeLa), human neuroblastoma cells (IMR32) and human promyelocytic leukaemia cells (HL60)		[55]
0.1–0.5% O_2_ for up to 72 h; 0.2% O_2_ for 48 h	HATs? TETs?		Human Burkitt’s lymphoma cells (P493-6), mouse embryonic fibroblasts (MEF) and mouse hepatoma cells (Hepa 1–6)	PDK1 activation; blockage of PDH activity; reduced acetyl-CoA synthesis; widespread repression of RNA and mRNA synthesis	[56,57]
0.5% O_2_ for up to 48 h	KDM4C	ND	Human glioblastoma cells (SF188), human embryonic kidney cells (HEK293T), mouse embryonic fibroblasts (MEF), human neuroblastoma cells (SH-SY5Y), mouse bone marrow cells (32D) and mice fetal liver cells (FL5.12)	Increased 2-hydroxyglutarate (2HG) production; KDM4C inhibition, increased H3K9me3 levels	[58]
<0.5% O_2_ for up to 4 days	JMJD3, JARID1A and JARID1B	ND	Human lung fibroblasts (IMR-90)	JMJD3 activity is reduced, resulting in increased H3K27me3 levels in p16 promoter; JARID1A and JARID1B activity is also reduced, leading to increased H3K4me3	[59]
0.5% O_2_ for 16 h	H3ac	HIF-2α	Primary mice undifferentiated pleomorphic sarcoma cells	Reduced H3ac levels in *HIF-2**α* promoter leads to decreased expression	[60]
0.01% O_2_ for 48 h	H3K4me and H3K9ac	ND	Breast carcinoma cells (MCF-7)	Increased H3K4me1,2,3 and H4K9ac in *VEGF* promoter leads to transcriptional activation	[61]
	H3K4me2,3, H3K9me3 and H3K9ac, LSD1	HIF-independent		Reduced H3K4me2,3, increased H3K9me3 and decreased H3K9ac levels in *BRCA1* and *RAD51* promoters with a consequent decreased expression	
1% O_2_ for up to 72 h	JMJD1A	ND	Hepatocellular carcinoma cells (PLC, HuH7, and HepG2)	*JMJD1A* silencing during hypoxia leads to reduced *N-cadherin* and *Twist* levels and increased *E-cadherin* levels	[62]
0.5% O_2_ for up to 24 h	G9a	HIF-1α-independent	Human lung carcinoma cells (A549), HEK293 and mouse embryonic stem cells (MES)	Higher H3K9me2 levels mediated in part by G9a in *MLH1* promoter region, decreasing its expression	[63]
<10 ppm O_2_ or desferrioxamine mesylate treatment for up to 48 h	Histone acethylation	ND	Mouse fibroblasts (3340) and human cervix adenocarcinoma cells (HeLa)	Decreased MLH1 and PMS2 levels; increased mutation frequency	[64]
1% O_2_ for 24 h	JMJD1A, JMJD2B and JMJD2D	ND	Murine macrophages (RAW264.7)	Increased H3K9me in *MCP-1*, *CCR1* and *CCR5*, reducing their expression	[65]
0.5% O_2_ for 24 h	HDAC2	ND	Human cervix adenocarcinoma cells (HeLa)	HDAC2 recruitment by NF-κB leads to *MCP-1* downregulation	[66]

Legend: ND, not determined.

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
