# Peer review of "Regulation Is in the Air: The Relationship between Hypoxia and Epigenetics in Cancer"

_cells, 2019, doi:10.3390/cells8040300_

Round 1

Reviewer 1 Report

In this manuscript Camuzi and colleagues describes the crosstalk between low oxygen and epigenetics, and how this relationship contributes to the malignant phenotype.  The review in general is easy to follow, but would benefit from language editing to improve the English.  In addition, I have some specific suggestions that would benefit the article.

1.     The section describing the Jumonji Domain Containing proteins would benefit from being expanded.  There is a large amount of work on this subject that preceeded the references mentioned that would benefit from being referenced e.g

https://www.ncbi.nlm.nih.gov/pubmed/18984585

https://www.ncbi.nlm.nih.gov/pubmed/18713068

https://www.ncbi.nlm.nih.gov/pubmed/18538129

https://www.ncbi.nlm.nih.gov/pubmed/19858293

Alternatively, direct the reader to a review of this area of research.

2.      Figure 1 is and excellent and concise review of the manuscript, but the Greek letters need to be added to some of the HIF1alphas and HIF2alphas.

3.     Within the immune modulation section it may be beneficial to indicate that both HIF1alpha and HIF1beta are direct targets of NF-kB, contributing to cross talk between these pathways.  

https://www.ncbi.nlm.nih.gov/pubmed/18432192

https://www.ncbi.nlm.nih.gov/pubmed/18393939

https://www.ncbi.nlm.nih.gov/pubmed/21298084

Author Response

In this manuscript Camuzi and colleagues describes the crosstalk between low oxygen and epigenetics, and how this relationship contributes to the malignant phenotype.  The review in general is easy to follow, but would benefit from language editing to improve the English.  In addition, I have some specific suggestions that would benefit the article.

AUTHORS: We thank the reviewer for all the valuable comments and suggestions. Everything was considered in the new version of the manuscript and we are convinced the alterations helped improving it. Also, the language was rechecked and some sentences we rephrased.

 1.     The section describing the Jumonji Domain Containing proteins would benefit from being expanded.  There is a large amount of work on this subject that preceeded the references mentioned that would benefit from being referenced e.g

https://www.ncbi.nlm.nih.gov/pubmed/18984585

https://www.ncbi.nlm.nih.gov/pubmed/18713068

https://www.ncbi.nlm.nih.gov/pubmed/18538129

https://www.ncbi.nlm.nih.gov/pubmed/19858293

Alternatively, direct the reader to a review of this area of research.

AUTHORS: The suggested references were added and the section regarding JMJD proteins was extended in the manuscript. The alterations can be found in lines 315-329.

“Histone demethylases from the Jumonji Domain-Containing (JMJD) family of proteins have also been suggested to play a role in the interplay between hypoxia and epigenetics. Wellmann et al. [53] showed in human embryonic kidney cells (HEK-293) and human microvascular endothelial cells (HMEC-1) that hypoxia and hypoxia mimetic agents induce JMJD1A expression, which is abrogated after HIF-1A silencing [53]. The authors also showed that JMJD1A promoter harbors a hypoxia responsive element, enabling its induction by HIF-1α both in vitro and in vivo. HIF-1α is also able to bind to JMJD2B promoter, inducing its expression [54]. The regulation of JMJD proteins by hypoxia was also evaluated in other human cancer cell lines (U2OS, MCF7, HeLa, IMR32 and HL60), confirming the induction of JMJD1A and JMJD2B [55]. JMJD2B upregulation by HIF-1α during hypoxia has been linked to the modulation of hypoxic gene expression, and to increased cell proliferation [51]. Furthermore, the regulation of a subset of hypoxia-inducible genes, including ADM and GDF15, has been shown to be dependent on JMJD1A in renal cell and colon carcinoma cell lines, reinforcing the connection between hypoxia, HIF-1α and JMJD-mediated chromatin remodeling [52].”

2.      Figure 1 is and excellent and concise review of the manuscript, but the Greek letters need to be added to some of the HIF1alphas and HIF2alphas.

AUTHORS: The Figure was revised and we believe all Greek letters are now included.

3.     Within the immune modulation section it may be beneficial to indicate that both HIF1alpha and HIF1beta are direct targets of NF-kB, contributing to cross talk between these pathways. 

https://www.ncbi.nlm.nih.gov/pubmed/18432192

https://www.ncbi.nlm.nih.gov/pubmed/18393939

https://www.ncbi.nlm.nih.gov/pubmed/21298084

AUTHORS: The suggested references were added and a sentence indicating HIF1alpha and HIF1beta as targets of NF-kB was added in the manuscript. The alterations can be found in lines 532-533.

“NF-kB also directly regulates HIF-1α and HIF-1β gene expression, resulting in upregulation of several hypoxia-like genes [127-129].”

Reviewer 2 Report

In this manuscript Camuzi et al. and colleagues describes the crosstalk between hypoxia and epigenetic changes in cancer. The review is generally well written and readable.

 I have some suggestions that need to be taken into account:

1) In the description of the hypoxia inducible factors (line 46-52) the authors should specify that the HIF-3α subunit lacks the C-terminal transactivation domain.

2) Line 65-70: VHL can also phosphorylate HIF-3α

3) Line 110-126: Since DNA methylation is a reversible process the authors should taken into account to add a clarifying figure to this paragraph.

4) Line 176: a reference is missing after “shorter survival”

5) Section “Hypoxia and DNA methylation” as well as section “Hypoxia and histone modification”: the authors should consider to add 2-3 clarifying sentences to the end of each section why DNA methylation or histone modifications regulated by hypoxia are important regarding to cancer.

6) Epigenetic changes regarding to hypoxia are very complex and affecting a large number of genes in many different cell types. Therefore, I would recommend that the authors taken into account to add a table which includes the major epigenetic players in response to hypoxia, the cell type, the system etc. in order to make the review easily understandable for the reader.

7) I recommend the authors to change the title. From my point of view, the phrase “regulation is in the air” is misleading.

8) I recommend the authors to carefully recheck the language of the manuscript and rephrase sentences where needed.

Author Response

In this manuscript Camuzi et al. and colleagues describes the crosstalk between hypoxia and epigenetic changes in cancer. The review is generally well written and readable.

AUTHORS: We thank the reviewer for all the valuable comments and suggestions. Everything was considered in the new version of the manuscript and we are convinced the alterations helped improving it.

 I have some suggestions that need to be taken into account:

1)      In the description of the hypoxia inducible factors (line 46-52) the authors should specify that the HIF-3α subunit lacks the C-terminal transactivation domain.

AUTHORS: The information was added in the manuscript, lines 52-53.

“HIF-1α and HIF-2α have both N-terminal and C-terminal TADs, while HIF-3α has only the N-terminal TAD [14].”

2) Line 65-70: VHL can also phosphorylate HIF-3α

AUTHORS: The information was added in the manuscript, lines 65-70.

“Under normoxic conditions, the hypoxia pathway is inactivated by post-translational modifications. So, in the presence of normal oxygen levels, α subunits proline residues 402 and 564 (HIF-1α), 405 and 531 (HIF-2α), and 490 (HIF-3α) are hydroxylated by the prolyl hydroxylase enzymes (PHDs). Proline modifications are then recognized by Von Hippel-Lindau (VHL) proteins that are substrates for the E3 ubiquitination complex. Thus, α subunits are rapidly degraded by the proteolytic pathway [17,18].”

3) Line 110-126: Since DNA methylation is a reversible process the authors should taken into account to add a clarifying figure to this paragraph.

AUTHORS: A new Figure 1 was added to the manuscript, clarifying DNA methylation and demethylation as well as the establishment and erasure of histones post-translational modifications.

4) Line 176: a reference is missing after “shorter survival”

AUTHORS: The corresponding references were added in the manuscript.

5) Section “Hypoxia and DNA methylation” as well as section “Hypoxia and histone modification”: the authors should consider to add 2-3 clarifying sentences to the end of each section why DNA methylation or histone modifications regulated by hypoxia are important regarding to cancer.

AUTHORS: Clarifying sentences were added to the end of each section.

Hypoxia and DNA methylation (lines 238-244): “Based on these data, we may conclude that the hypoxic phenotype is at least partially mediated by DNA methylation alterations, depending on both the modulation of SAM’s availability and the regulation of enzymes involved in DNA methylation and demethylation by HIFs. In the context of cancer, this link contributes to the establishment of a recurrent tumor epigenotype, involving both global DNA hypomethylation and tumor suppressor hypermethylation. Therefore, DNA methylation alterations induced by hypoxia may play a pivotal role in tumorigenesis.”

Hypoxia and histone modification (lines 361-373): “The interplay between hypoxia, HIFs and histone modifications seems to be even more complex than that with DNA methylation. Indeed, the data gathered here show that histone modifiers are capable of modulating HIFs stability, having a direct impact on hypoxic signaling, but HIFs are also capable of modulating the expression of histone modifiers. Furthermore, metabolism intermediates, highly sensible to O2 levels, are cofactors of these enzymes, and HIFs may also exert its transcriptional activities through the interaction with chromatin remodelers. In cancer, the expression of different histone-modifying enzymes has been shown to be dysregulated, what can be mediated at least in part by HIFs, and has an impact on tumor phenotype and prognosis. Also, some hypoxia-associated tumor transcriptional programs seem to be acquired by the cooperation of HIFs and histone modifiers. Although much is left to be clarified, these observations suggest a strong connection between hypoxia and this epigenetic mechanism in the establishment of cancer hallmarks.”

6) Epigenetic changes regarding to hypoxia are very complex and affecting a large number of genes in many different cell types. Therefore, I would recommend that the authors taken into account to add a table which includes the major epigenetic players in response to hypoxia, the cell type, the system etc. in order to make the review easily understandable for the reader.

AUTHORS: A new table summarizing all data described in the review was added.

7) I recommend the authors to change the title. From my point of view, the phrase “regulation is in the air” is misleading.

AUTHORS: We chose the title to allude to how oxygen levels (“air”) may impact on epigenetic mechanisms, taking as reference the song “Love is in the air” by John Paul Young. But if the reviewer considers this is not clear, we could shorten the title (“The relationship between hypoxia and epigenetics in cancer”).

8) I recommend the authors to carefully recheck the language of the manuscript and rephrase sentences where needed.

AUTHORS: The language was rechecked and some sentences we rephrased.

Round 2

Reviewer 2 Report

The authors have addressed my main concerns and improved the quality of the review article.